# The Nebulous Association between Cognitive Impairment and Falls in Older Adults: A Systematic Review of the Literature

**DOI:** 10.3390/ijerph20032628

**Published:** 2023-02-01

**Authors:** Victoire Leroy, Valérie Martinet, Oumashankar Nunkessore, Christel Dentel, Hélène Durand, David Mockler, François Puisieux, Bertrand Fougère, Yaohua Chen

**Affiliations:** 1Division of Geriatric Medicine, Tours University Hospital, 37000 Tours, France; 2EA 7505 (Education, Ethics, Health), University of Medicine of Tours, 37000 Tours, France; 3Memory Clinic, Tours University Hospital, 37000 Tours, France; 4Department of Geriatrics, Saint-Pierre Hospital, ULB, 1000 Brussels, Belgium; 5Department of Geriatrics, Dunkerque Hospital, 59240 Dunkerque, France; 6Neurology Practice, 67170 Brumath, France; 7Department of Neurology, Hautepierre Hospital, Strasbourg University Hospital, 67200 Strasbourg, France; 8Medical Library, Trinity Centre for the Health Sciences, St James’ Hospital, D08 W9RT Dublin, Ireland; 9Department of Gerontology, Lille University Hospital, 59000 Lille, France; 10EA2694, Lille University, 59000 Lille, France; 11INSERM UMR-S 1172, Vascular and Degenerative Cognitive Disorders, University of Lille, 59000 Lille, France; 12Global Brain Health Institute, Trinity College Dublin, D02 PN40 Dublin, Ireland

**Keywords:** falls, mild cognitive impairment, gait disorders, cognitive impairment

## Abstract

Background: In older people, dementia is a well-established risk factor for falls. However, the association and the causal relationship between falls and the earlier stages of cognitive impairment remains unclear. The purpose of the study was to review the literature data on the association between falls and cognitive impairment, no dementia, including Mild Cognitive Impairment. Methods: According to PRISMA guidelines, we searched five electronic databases (EMBASE, Web of Science, Medline, CINAHL, and PsychINFO) for articles published between January 2011 and August 2022 on observational studies of older people with a cognitive assessment and/or cognitive impairment diagnosis and a recording of falls. Their quality was reviewed according to the STROBE checklist. Results: We selected 42 of the 4934 initially retrieved publications. In 24 retrospective studies, a statistically significant association between falls and cognitive status was found in only 15 of the 32 comparisons (47%). Of the 27 cross-sectional analyses in prospective studies, only eight (30%) were positive and significant. We counted four longitudinal analyses, half of which suggested a causal relationship between falls and cognitive impairment. The investigational methods varied markedly from one study to another. Conclusion: It is still not clear whether falls are associated with cognitive impairment, no dementia. Data in favor of a causal relationship are scarce. Further studies are needed to clarify their relationship.

## 1. Introduction

The World Health Organization has defined a fall as an unintentional event in which a person falls to the ground or to any other lower level, excluding an intentional change in position [1]. In people over 65, the annual prevalence of falls can be as high as 28% or 35% [1,2]. Falls are associated with significant health consequences in terms of bone fractures, hospital admissions, and institutionalization [3,4]. Falls are therefore associated with a significant cost, which was estimated to be EUR 31 billion a year in the USA in 2015 [5]. An effective fall-prevention strategy must, therefore, identify and address all the risk factors in older people.

Age and dementia are strong, independent risk factors for falls [1,6]. In the Diagnostic and Statistical Manual of Mental Disorders, Fifth Edition (DSM-5) [7], major cognitive impairment—or dementia—is defined as an objective cognitive impairment and a loss of autonomy. The neurological processes involved in the central nervous system’s control of balance and gait are complex. Severe deterioration of these neurological processes in cases of dementia (with resulting impairments in executive [8], visuospatial [9], and attentional [10] functions) might explain this association.

Prior to the dementia stage, mild cognitive impairment (MCI) might also be a risk factor for falls. If it is precisely defined according to Petersen criteria [11] and DSMV [7] as an objective cognitive decline with the maintenance of functional independence, this varies from one literature source to another. Some experts have used specific cognitive scores (e.g., the Clinical Dementia Rating (CDR)) to define MCI [12], whereas others have used more general cognitive screening tools (e.g., the Mini Mental State Examination (MMSE)) [13]. Moreover, the cut-offs used vary greatly. More generally, other terms (such as “cognitive impairment” and “cognitive impairment, no dementia”) are sometimes used in the literature but do not necessarily correspond to the criteria established for MCI [14] and may cover a broad spectrum of cognitive disorders [15]. Furthermore, it is not clear whether MCI or cognitive impairment, no dementia is associated with the prevalence of falls.

In a systematic review published in 2012 [16], Muir et al. assessed 26 prospective cohort studies of institutionalized or community-dwelling people aged 60 and over, with falls as the main outcome and a cognitive assessment at inclusion. The researchers concluded that the global cognitive decline (i.e., in all stages) was associated with falls in general and with serious falls in non-institutionalized adults. More specifically, impaired executive function was associated with a greater risk of falling. However, the early stages of cognitive impairment were not specifically compared.

Since 2012, additional data on the potentially causal association between the risk of falling and cognitive impairment, no dementia have been published, especially in patients with MCI. If it is strongly established that patients with non-major cognitive impairment, such as cognitive impairment, no dementia, and/or MCI, have more gait disorders than cognitively unimpaired counterparts [17,18], their association with falling remains unclear. When compared, the prevalence of falls is indeed not significantly higher in these patients, but it was not their main objective [19,20].

Therefore, we sought to describe the results of an up-to-date systematic review of studies probing the association between cognitive impairment, no dementia and falls in older adults. More specifically, we sought to (i) describe the prevalence of falls with cognitive status in older people without dementia and (ii) assess their putative association probed in cross-sectional studies and the putative causal relationship probed in longitudinal studies.

## 2. Materials and Methods

### 2.1. Research Strategy and Information Sources

We used a logical combination of keywords related to falls, cognitive impairment, and older people (Appendix A) to search the EMBASE, Web of Science, MEDLINE, Cumulative Index to Nursing and Allied Health Literature (CINAHL), and PsycINFO literature databases for articles published between January 2011 and August 2022. The search strategy was reviewed by a qualified librarian (DM). Our keyword-based search yielded 10,458 hits. We extracted the list of publications into an online reference management tool (EndNote^®^, version 20, Clarivate, Philadelphia, PA, USA) to eliminate duplicates and then used a collaborative online tool (Covidence^®^, Melbourne, Australia) for the following steps. This systematic review indeed followed the PRISMA guidelines [21] and has been registered at the PROSPERO (International Prospective Register of Systematic Reviews) (CRD42022363363).

### 2.2. Study Eligibility Criteria

Four independent reviewers (VL, VM, HD, CD) read all the abstracts once and read all the subsequently selected full-text articles twice. We applied the following study inclusion criteria: participants aged 65 and over; residences at home or in an institution; a prospective cross-sectional cohort, retrospective cohort or population-based studies; “falls” as the primary or secondary outcome (including “recurrent falls”), regardless of the data collection method; and a cognitive assessment or diagnosis of MCI at the baseline. Given the heterogeneity of the definitions of MCI in the literature, we did not restrict our search to studies that had applied the currently accepted criteria for this condition. We excluded the following types of study: literature reviews, conference papers, case reports, animal studies, PhD theses, qualitative studies, research protocols, randomized controlled trials, pilot studies, studies of hospitalized patients, studies of particular groups of patients (such as those with Parkinson’s disease, stroke, etc.), and studies of people with a diagnosis of dementia.

### 2.3. Procedure for the Collection and Inclusion of Articles

After the initial list of publications had been drawn up, we performed the following steps in accordance with the Preferred Reporting Items for Systematic Reviews and Meta-Analyses (PRISMA) guidelines [21]: screening of the titles and abstracts, with the application of the study inclusion and exclusion criteria; double-blind assessment of the selected full-text articles by two researchers; a data extraction step in which the researcher who included the article extracted the data. In the event of disagreement, a fifth independent reviewer (YC) decided whether to include or exclude the publication and specified the reasons for the decision. Each extraction was validated by a second researcher.

The following variables were extracted and plotted on a shared online spreadsheet (GoogleSheets^®^, Google, Dublin, Ireland): name of the first author, year of publication, design, study sample size, mean age of the participants, percentage of females in the study population, country, length of follow-up (for a prospective study), the participants’ place of residence (community or institution), the study’s objective, methods of cognitive assessment, the presence or absence of a diagnosis of MCI and the diagnostic criteria used in such a case, the percentage of the total study population with cognitive impairment, and the methods used to analyze and compare data (statistical tests, adjustment factors, control group, etc.). If applicable, the interpretation of neuropsychological tests followed authors’ recommendations.

We differentiated between retrospective studies and prospective studies, i.e., depending on how fall events were recorded. Next, when considering prospective studies, we differentiated between longitudinal designs and cross-sectional designs according to the analyses performed. If several analyses of the same data had been performed, only the most relevant analysis was extracted for our review.

We considered the results of the analyses to be significant if *p* < 0.05.

### 2.4. Assessment of the Quality of Reporting

We used the 34-item STrengthening the Reporting of OBservational studies in Epidemiology (STROBE) checklist to determine the reporting quality of each selected study [22]. Each item present in the publication was scored as one point. We noted the total number of checklist points as an absolute value and as a percentage of the maximum possible for the type of study in question. After a team meeting and a review of previous work in this field, we considered a percentage of 75% as a threshold for a fair description [23].

## 3. Results

### 3.1. Selection of Publications

After eliminating duplicates, we initially identified 4934 publications (Figure 1). In our first screening step, 1198 of these publications were found to be eligible. After an assessment of the full-text publications, 42 were included in the review.

### 3.2. Description

The 42 selected publications described 18 prospective studies and 24 retrospective studies. Four of the studies assessed people living in an institution, 37 assessed community-based participants, and one study did not specify the living setting. The participants’ mean age ranged from 71.3 [24] to 85.2 [25].

When considering all of the studies of community-dwelling populations aged 65 and over (regardless of differences in other inclusion criteria), the prevalence of falls ranged from 4.5% [26] to 79.7% [27]. Multiple falls (i.e., at least two) were reported for 4.4% [24] to 13.8% [28] of the study participants.

Only 22 studies (52%) had, as a main objective, compared fall prevalence and cognition.

For a clearer presentation of the results, we divided the reviewed publications into retrospective and prospective studies. In the 24 retrospective studies, falls were recorded retrospectively (Table 1). Two of them had a longitudinal design.

Of the 18 prospective studies, two had a longitudinal design and 16 had a cross-sectional design (according to the baseline variables; Table 2).

With regard to the study populations, 11 studies included participants with MCI, 19 included patients with cognitive impairment, no dementia, and 18 included patients with cognitive score data. As described in the two tables, some studies included several types of patients (see below) and/or applied several types of cognitive or scores: global scores such as MMSE or MoCA, and also neuropsychological tests, i.e., the clock-drawing test, word recall, digit span and symbol, Trail Making tests A and B, fluency, the Stroop test, similarities, and logical memory, which may be used to define cognitive impairment or as continuous variables. We therefore identified 37 cross-sectional comparisons for the retrospective studies and 27 for the prospective studies.

### 3.3. The Association between Falling and Cognition

When considering the 22 retrospective studies with cross-sectional analyses, 15 (38%) of the 37 comparisons revealed a significant association between cognition and falling. A cognitive status was significantly associated with falls in five (71%) of the seven comparisons with a clear definition of MCI and 6 (60%) of 10 comparisons concerning cognitive impairment, no dementia. A global cognitive score was significantly associated with falls in two (25%) of the eight studies. One of the four studies assessing, cross-sectionally, an executive score and falls (25%) found a significant association (Figure 2).

When considering the 18 prospective studies, we identified 16 with cross-sectional designs (relative to the baseline cognitive status or scores) including a total of 27 comparisons (Figure 2): four with MCI status, six with cognitive impairment, no dementia, and 17 with global or specific cognitive scores. One (25%) of the four studies that included patients with MCI found a significant, positive association. Two (33%) of the six studies that included patients with cognitive impairment, no dementia showed a significant association. Of the four studies comparing a global cognitive score with the fall incidence, one found a significantly worse score (the MMSE score) in fallers. Of the seven analyses of specific cognitive scores, four (57%) found a significant, positive association for executive function [61,65], attention [61], or processing speed [53]. Overall, a positive association was found in 23 (36%) of the 64 comparisons in cross-sectional studies.

When considering all comparisons between patients with MCI and falling, only 6 of the 11 studies (54%) found a significant association between them. Among the 16 cross-sectional analyses including patients with cognitive impairment, no dementia, only eight showed significant association (50%).

### 3.4. The Causal Relationship between Falling and Cognition

Only four studies conducted longitudinal analyses of the putative relationship between the incidence of falls and cognitive impairment or scores. The relationship was found to be causal in both analyses of the prospective studies, which compared cognitive impairment, no dementia and incidence of falls. In retrospective studies, none of the analyses showed significant results between falling and specific cognitive scores.

### 3.5. Definition and Assessment of Cognitive Status

MCI was defined according to the Petersen criteria in 6 of the 11 studies (54%). Two studies used an MMSE score to define MCI [25,45], and another study used a CDR score < 0.5 as a cut-off [51]. The last two studies used neuropsychological tests, with a cut-off of −1.5 standard deviations [42,64].

The definition of cognitive impairment, no dementia also varied from one study to another. The MMSE score [13] was most frequently used (47%) to define cognitive impairment, with a cut-off varying between 17 and 24.

Overall, 16 studies featured a neuropsychological assessment (40%). Short cognitive tests were also used, such as the MMSE [13] in 18 studies and the Montreal Cognitive Assessment (MoCA) [66] in four studies.

### 3.6. Recording of Falls

The method of recording falls varied from one study to another. Twenty-seven studies were based on self-reporting or caregiver reporting, in an interview with a physician. In eight studies, researchers gave a falls diary to the participant. All three studies of people living in an institution relied on observations by caregivers. Lastly, three studies did not specify the data collection method.

### 3.7. Quality Criteria

After an evaluation of the STROBE checklist, 14 of the 42 publications had a ratio of 75% or more and were considered to be of high quality.

## 4. Discussion

In the present systematic review, we found a significant, positive association in only, respectively, 54% and 50% of the cross-sectional comparisons between the falling and MCI group and between the falling and cognitive impairment, no dementia group. More precisely, in the included retrospective analyses with cross-sectional analyses, five (71%) of the seven comparisons with a clear definition of MCI, 6 (60%) of the 10 comparisons with cognitive impairment, no dementia, and two (25%) of the eight comparisons with global cognitive scores. When considering prospective studies with cross-sectional analyses, only 30% found a significant, positive association between cognition and falls: one (25%) of the four comparisons with a clear definition of MCI, two (33%) of the six comparisons with cognitive impairment, no dementia, and one (25%) of the four comparisons with global cognitive scores. The two prospective, longitudinal studies found a causal relationship between cognitive impairment and falling but not in the two retrospective, longitudinal analyses that used specific cognitive scores.

Our systematic review provided additional information to existing reviews and seemed to be contradictory to Muir et al.’s 2012 review, which found a strong association between traumatic falls and cognition (odds ratio [95% confidence interval] = 2.13 [1.56–2.90]) [16]. However, more recently, a meta-analysis found that global cognition was not significantly associated with falls [67]; this conclusion is more in line with our present results. Most studies and reviews estimated the risk of falling via associated markers, such as the presence of gait and balance disorders. For example, another review considered that the risk of falling was greater in patients with MCI or early onset major cognitive impairment because of changes in gait, balance, and fear of falling [68].

Thus, the result of our review tempers the hypothesis whereby non-major cognitive impairment is a risk factor for falls. This might be due to a high degree of heterogeneity with regard to the methods used to document falls, diagnose cognitive impairment/MCI, and perform cognitive assessments. This heterogeneity might also be related to the large body of literature data on this issue and raises the issue of poor inter-study comparability. Most of the included analyses were cross-sectional with heterogenous results between 25% and 66% of significant association. More specifically, the results concerning MCI and the risk of falling ranged from the absence of a clear association in some studies [60] to a strong association in others [26,51]. This heterogeneity might be also inherent to the concept of MCI. Indeed, patients with MCI are generally characterized by a cognitive impairment that is significant but does not influence their independence [69]. From 10 to 33% of adults with MCI will develop dementia in the following two years [70]. However, the fact that MCI appears to be reversible in a high proportion of cases (up to 46.5%) suggests the absence of a concomitant, active, neuropathologic process that might cause falls [71]. This marked variability in the course of MCI makes it difficult to interpret pathophysiological links with the risk of falling; depending on the study population, a link may or may not be found. The observed differences between studies might also be linked to the study duration, with stronger associations in longer studies. However, a 36-month prospective study did not show any association between falling and MCI status [64], whereas a similar 18-month study did [51].

The literature results were also heterogenous for comparisons of the prevalence or incidence of falls with measurements of precise cognitive functions in a standardized neuropsychological assessment. When data were available, executive dysfunction was the most frequently cognitive dysfunction associated with falling. However, only three of the eight studies that compared executive scores with falls found a significant, positive association. None of these eight studies were longitudinal. Some other studies showed that impairments in executive functions are particularly associated with a risk of falling in older adults, independent of the overall level of cognitive decline [8,62]. Abnormally slow gait in patients with MCI has been linked to impairments in executive functions and working memory [72]. These findings are in line with Muir et al.’s meta-analysis [16] and functional imaging studies [73].

There are several possible mechanistic explanations for the higher risk of falling that accompanies a decline in executive functions. Firstly, the decline might impede the ability to compensate for age-related changes in balance and walking [74]. Secondly, executive functions are required for the simultaneous performance of several tasks and, thus, integrating locomotion-related changes in the environment [75]. Thus, many studies of older people with falls have highlighted the latter’s poor performance in dual tasks (and especially those involving walking) [76,77]. This is well-illustrated by the association between future falls and performances in the “stop walking while talking test” [78].

We also reviewed a number of confounding factors assessed in some studies, including social engagement [42], caregiving [59], social isolation [38], brain imaging data [46], and muscle strength [47]. These studies illustrated (i) the complexity of the association between the risk of falling and cognitive impairment, and (ii) the interplay between the many risk factors for falls. Indeed, the fact that the risk of falling is always multifactorial emphasizes the value of multidisciplinary assessments [79]. Other risk factors described in the literature include personality traits, which can be significantly associated with falls even when cognitive disorders are absent [80].

Only two prospective, longitudinal analyses were conducted among the 42 articles included. Interestingly, both showed a positive and significant association between falls and cognitive impairment, no dementia, suggesting a causal relationship. However, only one was adjusted [50] and both compared cognitive impairment according to scores without precision for MCI [49,50]. There is a need of longitudinal studies for a better understanding of the relationship between MCI and fall incidence.

Our review had a number of strengths. Firstly, to make our review as exhaustive as possible, we considered a variety of cognitive concepts and ways of assessing the association between falls and cognitive disorders. Secondly, we adopted a systematic approach. Thirdly, we chose to include studies of people aged 65 and over, the age group most frequently assessed in the literature. The fall risk increases with age, although some studies of patients below the age of 65 have found a significant association between MCI and falls [81]. This highlights the difficulty of choosing an age cut-off. Fourthly, we included only studies with falling outcomes and excluded indirect markers, such as gait disorders, fear of falling, and standardized walking tests.

Our review also had some limitations. Firstly, 21 of the reviewed studies were not primarily designed to investigate the association between falls and cognitive impairment and so might have lacked power in this respect. Secondly, we chose to include non-prospective studies, the results of which might be influenced by recall bias, especially in populations of patients with cognitive impairment. Indeed, it has been shown that cognitively impaired patients are more likely to forget about events and have timing bias [82]. Thirdly, some of the analyses were only univariate, which again makes it difficult to compare study results. Hence, some results should be interpretated with caution. Fourthly, although we excluded studies of populations with a specific disease (e.g., Parkinson’s disease) and those with dementia, it is possible that some of the patients in the included studies did suffer from one of these conditions (e.g., when a patient had an MMSE score below 20 and no other details were given). Fifthly, we also excluded conference papers and PhD theses in order to avoid duplicates. This “grey literature” is nevertheless of value in systematic reviews and reduced publication bias. Hence, our review might have omitted reports of negative results. Lastly, if the included studies varied greatly according to our inclusion criteria, it also makes their comparison and presentation more complex.

## 5. Conclusions

In conclusion, we highlighted the high degree of heterogeneity in the literature on falls and cognitive impairment. For example, the study designs and the methods used to assess cognitive performance and to record falls differed from one study to another. This made it more difficult to compare study results. Moreover, our review highlighted differences in the studies’ conclusions about falls in older adults with cognitive impairment but who were not demented. Although falls are strongly associated with dementia, it is still not clear whether a significant association holds for MCI and other non-major cognitive conditions. Our results indicate the presence of a non-significant trend but do not enable us to rule out a coexistence.

Dedicated and prospective studies with standardized data collection methods are now required for a better understanding of the putative link between falls and non-major cognitive impairment.

## Figures and Tables

**Figure 1 ijerph-20-02628-f001:**
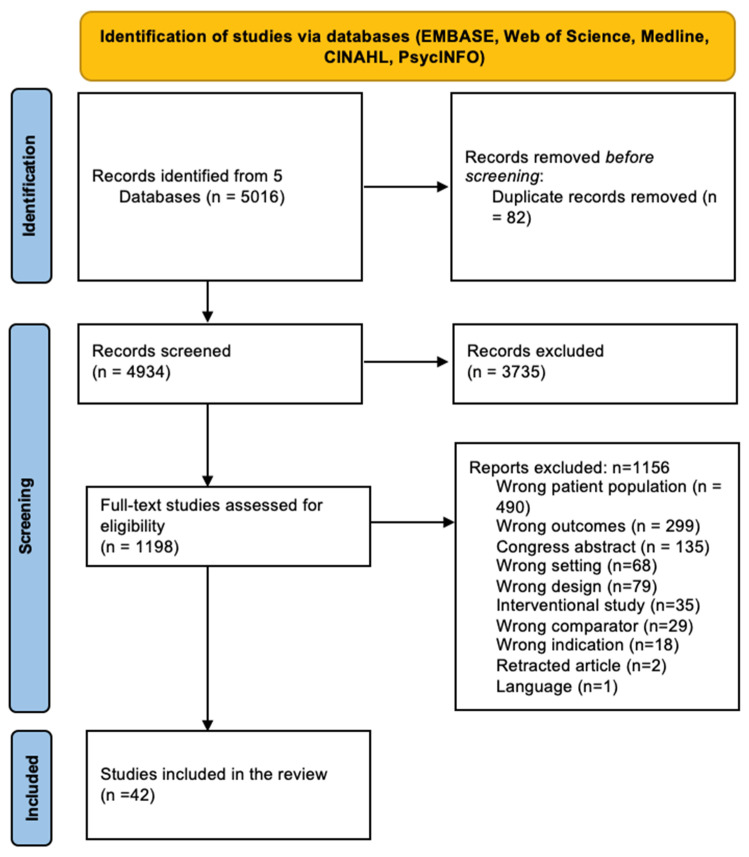
Flow chart: the literature screened for inclusion in the review.

**Figure 2 ijerph-20-02628-f002:**
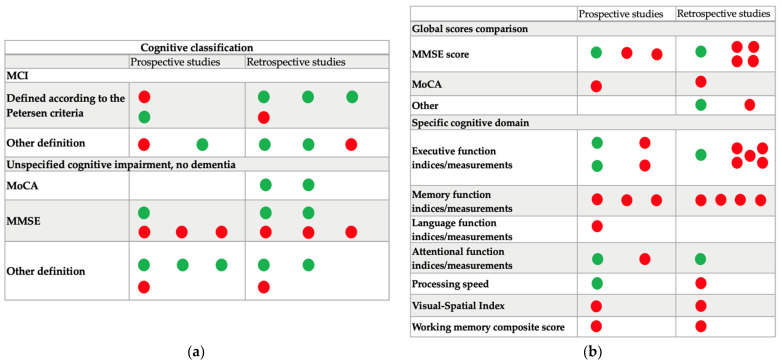
Summary of the association between falls and cognitive status (**a**) according to MCI and cognitive impairment, no dementia; (**b**) according to global and specific cognitive scores. Green point symbolizes a positive and significant association and red point symbolizes the absence of significant association. AD8: AD8 Dementia Screening Interview. MMSE: Mini Mental State Examination. MoCA: Montreal Cognitive Assessment. CDR: Clinical Dementia Rating. MCI: Mild Cognitive Impairment.

**Table 1 ijerph-20-02628-t001:** Retrospective studies assessing the association between cognition and falls.

Authors, Year, Country	N; Age *; Females ^†^	Fallers (%)	Cognitive Comparator: Definition	STROBE Score	Measurement of Effect Size (95% CI, *p)*
Longitudinal analysis					
Amini et al., 2022, USA [29]	1225; 74.2 (6); 57.7	from 19.5% to 25.8%	Executive and memory scores (*a*: Clock Drawing Test, *b*: Immediate and *c*: Delayed Word Recall Test)	50%	95% CI= *a*: [0.88–1.12]; *b*: [0.87–1.19]; *c*: [0.91–1.23]
Jayakody et al., 2022, USA [30]	522; 80.6 (5.3); 61.5	40.2%	Executive, attentional, memory & working memory scores (: *a*: TMT A & *b:* B, *c*: FAS, *d*: digit span, *e*: digit symbol, *f*: memory, *g*: Stroop test, *h*: category fluency)	73.3%	*p* = ***a*: 0.05**; *b*: 0.45; c: 0.51; *d*: 0.23; *e*: 0.37; f: 0.47; g: 0.89; h:0.18
Cross-sectional analysis					
Baixinho et al., 2018, Portugal ^‡^ [31]	204; NR; 71%	41.8%	Cognitive impairment according to MMSE without a specified cut-off	62.5%	*p* > 0.05
Diaz et al., 2020, Spain ^‡^ [25]	2849; 85.2 (NR); 68.3%	45.3%	- *a*: Mild cognitive impairment according to MMSE between 19 and 23 - *b*: Moderate cognitive impairment according to MMSE between 10 and 18	67.7%	*a*: *p* = 0.24; ***b*: *p* < 0.001**
Doi et al., 2015, Japan [24]	3400; 71.5 (5.2); 53.1%	4.4%	MCI according Petersen Criteria	75%	**95%CI = [1.03–2.37]**
Dokuzlar et al., 2020, Turkey [32]	682; 74.4 (8.5); 100%	31.5%	MMSE score	43.8%	*p* = 0.66
Ferrer et al., 2012, Spain [33]	328; NR; 61.6%	28.4%	MEC score	75%	*p* < 0.03
Halliday et al., 2018, Canada [34]	27;76.1 (3.3); 55.5%	44.4%	Executive and memory scores (*a*: MAT, *b*: digit symbol, *c*: letter series, *d*: similarities, *e*: vocabulary, *f*: recall)	67.7%	*p* = **a**: 0.41; *b*:0.23; c:0.69; **d: 0.004**; e:0.24; f:0.11
Kabeshova et al., 2014, France [35]	1760; 71.0 (5.1); 49.4%	19.7%	Cognitive impairment according to short-MMSE	76.7%	*p* = 0.38
Langeard et al., 2019, Canada [36]	26; 75.5 (NR); 88%	42.3%	- *a*: Cognitive impairment (MoCA < 26) - *b*: MoCA and *c*: MMSE scores - Cognitive composite scores (d: EF, e: memory, f: processing speed, g: visuospatial skills)	75.8%	*p* = - ***a*: 0.04**. - *b*: 0.61; c: 0.07 - d:0.96; e: 0.56; f: 0.95; g: 0.07
Lauretani et al., 2018, Italy [37]	451; 82.1 (6.8); 66.7%	54.3%	MMSE score	75%	***p* < 0.001**
Lee et al., 2011, Taiwan [38]	173; 78.8 (6.8); 26.6%	24.2%	Cognitive impairment according to MMSE < 24	50%	*p* = 0.158
Li et al., 2020, USA [39]	670; 77.7 (5.6); 65.1%	73.4%	Cognitive impairment according to MoCA < 23	73.9%	***p* < 0.001**
Merlo et al., 2012, Switzerland [28]	130; 79 (6): 60%	34.6%	MMSE score	71.4%	*p* = 0.13
Montero-Odasso et al., 2012, Canada [40]	68; 73.3 (NR); 66.2%	41.2%	MCI according to Petersen criteria	69%	*p* = 0.01
Muir et al., 2012, France [41]	4481; 71.8 (5.4); 47.6%	28.1%	Abnormal Executive function according to 1 or more errors in CDT	71%	*p* = 0.008
Quach et al., 2019, USA [42]	430; 76.6 (7); 68%	42%	MCI according to < 1.5 SD in ≥ 2 tests in NPA	57.6%	***p* < 0.0001**
Shirooka et al., 2017, Japan [43]	470; 73.6 (5.2); 70%	19.4%	Objective Cognitive Decline according to MMSE ≤ 23	54.8%	***p* = 0.02**
Smith et al., 2020, China, Ghana, India, Mexico, Russia, South Africa [26]	13623; 72.3 (10.9); 54.4%	4.5%	MCI according to Petersen Criteria	77%	**95% CI= [1.12–2.07]**
Tsutsumimoto et al., 2018, Japan [44]	10,202; 73.7 (5.5); 51.5%	NR	Cognitive impairment according to score < 1.5 SD in each of 3 neuropsychological tests	66.7%	***p* = 0.02**
Woo et al., 2017, Singapour [45]	385; NR; 63.9%	27.8%	- MCI according to MMSE score between 18 and 23 - -severe cognitive impairment according to MMSE < 17	73.1%	**95% CI= [1.08–3.25]** 95% CI= [0.73–2.59]
Yamada et al., 2013, Japan [46]	31; 78.9 (7.3); 74.2 %	32.2%	- MCI according to Petersen Criteria - MMSE score - executive scores from 4 neuropsychological tests (*a*: word and *b*: letter fluency, *c*: CDT, *d*: TMA)	73.1%	*p* = 0.32 *p* = 0.59 *p*= **a: 0.002**; b: 0.88; **c: 0.01**; d:0.53
Yang et al., 2018, Taiwan [47]	1067; 76.4 (6.0); 58.9%	15.1%	cognitive impairment according to CDR ≥ 2	71.4%	***p* = 0.05**
Zhou et al., 2022, China [48]	660; NS (sub-group)	21.4%	cognitive function as continuous variable	71.0%	***p* = 0.001**

N: number of participants; * Age is described as mean (SD); ^†^: the proportion of female participants is reported in percentage. ^‡^: studies including patients living in institution. CI: (Confidence Interval) CDR: Clinical Dementia Rating; EF: Executive Function; MCI: Mild Cognitive Impairment; MEC: Mini Examen Cognitivisco; MMSE: Mini Mental State Examination; MoCA: Montreal Cognitive Assessment; NPA: Neuropsychological Assessment; NR: Not reported; SD: Standard Deviation.

**Table 2 ijerph-20-02628-t002:** Prospective studies assessing the association between falls and cognitive decline.

Authors, Year, Country	N; Age *; Proportion of Females ^†^	Length of Follow-Up	Fallers (%)	Cognitive Comparator: Definition	STROBE Score	Measurement of Effect Size (95% CI, *p)*
Longitudinal analysis						
Ge et al., 2021, USA [49]	6000; NR; 56.4%	6 years	30.1% (baseline)	Cognitive impairment (according to score < 1/15th on memory and/or executive tests; or physician diagnosis; or AD8 ≥ 2)	81.8%	***p* = 0.04**
Ma et al., 2021, China [50]	965;74.9 (3.7); 53.2%	3 years	10.6%	Cognitive impairment (according to 20% of the lowest HDS-R scores and HDS-R > 10)	80.7%	***p* = 0.01**
Cross-sectional analysis						
Adam et al., 2021, USA [51]	2705; 78.5 (3.2); 45%	18 months	18.4%	MCI (according to CDR = 0.5 and < 10th percentile in ≥ 2 tests in NPA)	84.8%	***p* < 0.01**
Chantanachai et al., 2022, Australia [52]	266; 78.8 (NC); 45	12 months	39.8%	Global, executive & memory (*a*: MMSE, TMT *b*: A and *c*: B, *d*: FAS, *e*: digit symbol, *f*: logical memory) scores among patients with MCI according to Petersen Criteria	76.7%	*p*= a: 0.51, b: 0.68, c: 0.87, d: 0.93, e: 0.65, f: 0.85
Davis et al., 2017, Canada [53]	288; 81.5 (6.5); 69%	12 months	58%	- Cognition/processing speed composite score - Cognition/working memory composite score	75.7%	***p* < 0.01** *p* > 0.05
Dixe et al., 2021, Portugal ^‡^ [54]	204; NC; 71.1	12 months	41.7%	Cognitive decline according to MMSE score	58.1%	*p* = 0.78
Delbaere et al., 2012, Australia [55]	419; 77.8 (4.6); 53.9%	12 months	33.7%	- MCI according to Petersen criteria - cognitive (*a*: executive, *b*: memory, *c*: language, *d*: attention composite) scores	93.8%	95%CI = [0.90–2.63] 95%CI = a: [0.97–1.55], b: [0.71–1.13], c: [0.92–1.50], d: [0.81–1.28]
De Vries et al., 2013, Netherlands [56]	1509; 75.6 (NR); 51.8%	12 months	31.0%	Cognitive impairment according to MMSE ≤ 24	72.7%	95%CI = [0.68–1.82]
Franse et al., 2017, Europe multicentric [57]	18596; 74.1 (NR); 55.8%	2 years	8.4%	Cognitive impairment according to composite score < 1/10th	72.7%	*p* varying between > 0.05 and **<0.001** depending on the country
Gillain et al., 2019, Belgium [58]	96; 71.3 (5.4); 50%	2 years	36.5%	MoCA score	81.8%	*p* = 0.62
Hoffman et al., 2017, USA [59]	4528; 76.4 (NR); 58%	24 months	48.6%	Cognitive impairment according to TICS ≤ 8	65.6%	*p* = 0.36
Makizako et al., 2013, Japan [60]	42; 75.6 (6.3); 42.9%	12 months	26.2%	MMSE score among patients withMCI according to Petersen criteria	57.6%	*p* = 0.11
Mirelman et al., 2012, Israel [61]	256;76.5 (4.5);61%	5 years	71%	- MMSE score - *a*: Executive, *b*: attention, *c*: visual-spatial, *d*: memory composite scores	87.9%	*p* = 0.59 *p*= ***a*: 0.02, *b*: 0.002**, *c*: 0.74, *d*: 0.82
Pelaez et al., 2015, Spain ^‡^ [27]	74; 84(7); 79.7%	20 months	79.7%	MMSE score	54.5%	**95%CI = [1.03–1.40]**
Ranaweera et al., 2013, Sri Lanka [62]	1200; 71.4 (6.8); 57%	4 months	12.8%	Cognitive impairment according to MMSE < 24	55.9%	95%CI = [0.33–3.98]
Tchalla et al., 2014, USA [63]	765; 78.1 (5.4); 63.8%	5 years	69.9%	Cognitive impairment according to MMSE < 24	71.9%	***p* < 0.005**
Ward et al., 2019, USA [64]	365; NR; 67%	36 months	NR	(*a*: amnestic, *b*: non-amnestic et *c*: multiple domain) MCI according to score < 1.5 SD on 2 neuropsychological tests	65.6%	95%CI = **a: [1.10–2.61]**, b: [0.31–2.69], c: [0.25–1.91]
Zheng et al., 2012, Australia [65]	287; 77.8 (4.5); 53.7%	12 months	44.2%	TMT (A–B) score	69.6%	***p* = 0.03**

N: number of participants; * Age is described as mean (SD); ^†^: the proportion of female participants is reported in percentage; ^‡^: studies including patients living in institution. AD8: AD8 Dementia Screening Interview; MCI: Mild Cognitive Impairment; MMSE: Mini Mental State Examination; MoCA: Montreal Cognitive Assessment; NPA: neuropsychological assessment; NR: Not reported; TICS: Telephone Interview for Cognitive Status; TMT: Trail Making Test.

## Data Availability

Not applicable.

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
