# Peer review of "The Nebulous Association between Cognitive Impairment and Falls in Older Adults: A Systematic Review of the Literature"

_ijerph, 2023, doi:10.3390/ijerph20032628_

Round 1
Reviewer 1 Report
Strengths: This is an important topic with contradictory findings in the literature that would benefit from a systematic review. The search criteria and methods are clearly described in the methods and supplement and are an appropriate approach for a systematic review. The inclusion of the STROBE checklist is a strength. Figures 1 and 2 are well done and helpful. The authors clearly describe how falls were ascertained.
Major:
1 The title of the article is “How robust is the association between MCI and falls in older adults?” In the introduction, the authors use the definition of dementia from the DSM-V. The authors then describe how there are different definitions of MCI. MCI is clearly defined in the DSM-V as including subjective and objective cognitive decline in addition to continued ability to perform activities of daily living. The authors then state that their goal is to look at the association between “cognitive impairment, no dementia and falls in older adults”. There is a conflict here. “Cognitive impairment, no dementia” is not technically MCI, and including this as the definition for MCI further confuses our understanding of the association between MCI and falls. My recommendation is to restrict the inclusion of articles to those that adhere to a consistent, established definition of MCI- including objective and subjective measures of cognition, and a measure of activities of daily living. If the authors do not do this, I strongly recommend they change the title of their article to use the term “cognitive impairment” instead of MCI and do not refer to their review as assessing the association between MCI and falls. I recognize that there are very few articles that include an assessment of MCI that meets established criteria, but if the stated objective is to assess MCI and falls, it is necessary to restrict the analyses to articles that do meet these criteria in order to progress and refine our understanding of the relationship between MCI and falls. In the results section and the discussion section, there is some distinction made between the results from studies with clearly defined MCI, and those with cognitive impairment, but sometimes these studies are grouped together. If the authors are going to include cognitive impairment, not specifically MCI, and falls, they need to explain this more clearly in the title and introduction and separate the results and discussion of results into MCI and falls, and cognitive impairment and falls. This is a significant limitation.
2. The authors state that “only 22 studies (52%) had a main objective to compare fall prevalence and cognition”. Especially in this case, the authors should provide the measurements of effect size they are using from each study to determine whether there was a significant association between cognition and falls. This can be listed in Tables 1 and 2. The authors state multiple times throughout the manuscript that there was an association between cognition and falls in several studies but do not provide any evidence to support this. It is not possible to accurately assess the authors' conclusions and the quality of this systematic review without this information. This is a significant limitation.
3. The authors need to further describe the use of the terms “prospective” and “retrospective” when being applied to cross-sectional analyses. This is confusing. Typically, a study cannot be both “prospective or retrospective” and cross-sectional.
4. Please double check that the criteria listed under “comparison” are being consistently described in the results section.
5. In the discussion section, please distinguish between MCI and falls and cognitive impairment and falls. In the results the authors report that in 71% of articles that reported a clear definition of MCI, there was a positive association between MCI and falls. This seems like a strong association, however, this is not the way it is described in the discussion. Additionally suddenly using the term mild-to-moderate cognitive impairment in the discussion is confusing.
Minor:
1. Please do not use the acronym “MCI” in the title, use Mild Cognitive Impairment instead.
2. Tables 1. And 2. – The inclusion of the MCI criteria is very helpful; however the column heading “comparison” is not informative and is somewhat confusing.
3. Line 222 “ This section…” should be deleted.
4. In the discussion section “MCI status increased the risk of gait disorders and thus of falling” this needs a citation and more context. The authors did not look at this in their review, so it needs to be cited from other sources. In general, this paragraph seems out of place and the conclusions in it do not seem related to previous statements in the paragraph
5. Line 304 “Both resulted significantly”, change the phrasing of this
6. Line 314: “Fourthly, in order….” This sentence/ conclusion does not make sense.
7. Line 335, omit “data”, from “literature data”
Author Response
Strengths: This is an important topic with contradictory findings in the literature that would benefit from a systematic review. The search criteria and methods are clearly described in the methods and supplement and are an appropriate approach for a systematic review. The inclusion of the STROBE checklist is a strength. Figures 1 and 2 are well done and helpful. The authors clearly describe how falls were ascertained.
We deeply thank the reviewer 1 for his/her comments and recommendations. We are grateful for the opportunity to re-submit our manuscript with corrections. We have addressed each comment (in black Times New Roman font) in turn and present our responses in blue Times New Roman font.
Major:
- The title of the article is “How robust is the association between MCI and falls in older adults?” In the introduction, the authors use the definition of dementia from the DSM-V. The authors then describe how there are different definitions of MCI. MCI is clearly defined in the DSM-V as including subjective and objective cognitive decline in addition to continued ability to perform activities of daily living. The authors then state that their goal is to look at the association between “cognitive impairment, no dementia and falls in older adults”. There is a conflict here. “Cognitive impairment, no dementia” is not technically MCI, and including this as the definition for MCI further confuses our understanding of the association between MCI and falls. My recommendation is to restrict the inclusion of articles to those that adhere to a consistent, established definition of MCI- including objective and subjective measures of cognition, and a measure of activities of daily living. If the authors do not do this, I strongly recommend they change the title of their article to use the term “cognitive impairment” instead of MCI and do not refer to their review as assessing the association between MCI and falls. I recognize that there are very few articles that include an assessment of MCI that meets established criteria, but if the stated objective is to assess MCI and falls, it is necessary to restrict the analyses to articles that do meet these criteria in order to progress and refine our understanding of the relationship between MCI and falls.
We understand and agree with this point. Indeed, cognitive impairment, no dementia cannot be defined as MCI according to DSM V and Petersen criteria.
We showed in our work that the existing literature is limited by two points: on the one hand, the term MCI that could be used by some authors was not always clearly defined in accordance with the DSMV criteria; and on the other hand, many articles referred to a "non-major" cognitive impairment because it did not match the framework of dementia. Therefore, we changed our title and methodology to be exact, and respecting the verbatim proposed by the authors of each article. We agree that this makes the included articles more heterogeneous, but we believe it is closer to the clinical reality to show the link between falls and non-major cognitive decline.
We clarified those points in our rational:
- In the title: “The nebulous association between cognitive impairment and falls in older adults : a systematic review of the literature”
- In the abstract: “ In older people, dementia is a well-established risk factor for falls. However, the association and the causal relationship between falls and earlier stage of cognitive impairment remains unclear. The purpose of the study was to review the literature data on the association between falls and cognitive impairment, no dementia, including Mild Cognitive Impairment » (lines 22-25) and conclusion : “Dedicated and prospective studies with standardized data collection methods are now required for a better understanding of the putative link between falls and cognitive impairment” (lines 361-362). We also avoided to us the term MCI alone to specify it as including more generally cognitive impairment, no dementia.
- In the introduction:
- “major cognitive impairment -or dementia- is defined as an objective cognitive impairment and a loss of autonomy” (lines 50-51);
- “If it is precisely defined according to Petersen criteria [1] and DSMV [1] as an objective cognitive decline with the maintenance of functional independence, this varies from one literature source to another » (line 57-58).
In the results section and the discussion section, there is some distinction made between the results from studies with clearly defined MCI, and those with cognitive impairment, but sometimes these studies are grouped together. If the authors are going to include cognitive impairment, not specifically MCI, and falls, they need to explain this more clearly in the title and introduction and separate the results and discussion of results into MCI and falls, and cognitive impairment and falls. This is a significant limitation.
We first separated the included articles according to their designs, to clarify the association between falls and cognitive impairment, which might be causal in longitudinal studies. As you suggested, we highlighted results according to cognitive status only: “When considering all comparisons between patients with MCI and falling, only 6 of the 11 studies (54%) found a significant association between them. Among the 16 cross-sectional analyses including patients with cognitive impairment, no dementia, only 8 showed significant association (50%).” (lines 219-222) and further: “The relationship was found to be causal in both analyses of the prospective studies, which compared cognitive impairment, no dementia and incidence of falls” (lines 225-227)
- The authors state that “only 22 studies (52%) had a main objective to compare fall prevalence and cognition”. Especially in this case, the authors should provide the measurements of effect size they are using from each study to determine whether there was a significant association between cognition and falls. This can be listed in Tables 1 and 2. The authors state multiple times throughout the manuscript that there was an association between cognition and falls in several studies but do not provide any evidence to support this. It is not possible to accurately assess the authors' conclusions and the quality of this systematic review without this information. This is a significant limitation.
We added in tables & and 2 the p-values and/or 95% confidence intervals that were reported in the included studies. We also added the type of the cognitive test if applicable.
Table 1. Retrospective studies assessing the association between cognition and falls
|
Authors, year, country |
N; Age*; Females† |
Fallers (%) |
Comparison |
STROBE score |
Measurement of effect size (95% CI, p) |
|
Longitudinal analysis |
|||||
|
Amini et al., 2022, USA. |
1225 ; 74.2 (6) ; 57.7 |
from 19.5% to 25.8% |
- Cognitive scores (a: Clock Drawing Test, b: Immediate and c: Delayed Word Recall Test) |
50% |
95% CI= a : [0.88-1.12] ; b : [0.87-1.19] ; c : [0.91-1.23] |
|
Jayakody et al, 2022, USA |
522 ; 80.6 (5.3) ; 61.5 |
40.2% |
- Cognitive scores (a: TMT A & b: B, c: FAS, d : digit span, e : digit symbol, f : memory, g: Stroop test, h : category fluency) |
73.3% |
p= a : 0.05 ; b : 0.45 ; c : 0.51 ; d : 0.23 ; e : 0.37 ; f : 0.47 ; g : 0.89 ; h :0.18 |
|
Cross-sectional analysis |
|||||
|
Baixinho et al., 2018, Portugal ‡ |
204; NR; 71% |
41.8% |
Cognitive impairment according to MMSE without a specified cut-off |
62.5% |
p > 0.05 |
|
Diaz et al., 2020, Spain ‡ |
2849; 85.2 (NR); 68.3% |
45.3% |
- a: Mild cognitive impairment according to MMSE between 19 and 23 - b: Moderate cognitive impairment according to MMSE between 10 and 18 |
67.7% |
a: p=0.24; b: p<0.001 |
|
Doi et al., 2015, Japan |
3400; 71.5 (5.2); 53.1% |
4.4% |
-MCI according Petersen Criteria |
75% |
CI = [1.03-2.37] |
|
Dokuzlar et al, 2020, Turkey |
682; 74.4 (8.5);100% |
31.5% |
MMSE score |
43.8% |
p=0.66 |
|
Ferrer et al., 2012, Spain |
328; NR; 61.6% |
28.4% |
MEC score |
75% |
p < 0.03 |
|
Halliday et al., 2018, Canada |
27;76.1 (3.3); 55.5% |
44.4% |
Neuropsychological scores (a: MAT, b: digit symbol, c: letter series, d: similarities, e: vocabulary, f: recall) |
67.7% |
p= a:0.41; b:0.23; c:0.69; d:0.004; e:0.24; f:0.11 |
|
Kabeshova et al., 2014, France |
1760; 71.0 (5.1) ; 49.4% |
19.7% |
Cognitive impairment according to short-MMSE |
76.7% |
p=0.38 |
|
Langeard et al., 2019, Canada |
26; 75.5 (NR); 88% |
42.3% |
- a: Cognitive impairment (MoCA < 26) - b: MoCA and c: MMSE scores - Cognitive composite scores (d: EF, e: memory, f: processing speed, g: visuospatial skills) |
75.8% |
p= - a: 0.04. - b: 0.61; c: 0.07 - d:0.96; e: 0.56; f: 0.95; g: 0.07 |
|
Lauretani et al., 2018, Italy |
451; 82.1 (6.8); 66.7% |
54.3% |
MMSE score |
75% |
p<0.001 |
|
Lee et al., 2011, Taiwan |
173; 78.8 (6.8); 26.6% |
24.2% |
Cognitive impairment according to MMSE < 24 |
50% |
p=0.158 |
|
Li et al., 2020, USA |
670; 77.7 (5.6); 65.1% |
73.4% |
Cognitive impairment according to MoCA < 23 |
73.9% |
p<0.001 |
|
Merlo et al., 2012, Switzerland |
130;79 (6): 60% |
34.6% |
MMSE score |
71.4% |
p=0.13 |
|
Montero-Odasso et al., 2012, Canada |
68 ; 73.3 (NR) ; 66 .2% |
41.2% |
MCI according to Petersen criteria |
69% |
p=0.01 |
|
Muir et al., 2012, France |
4481; 71.8 (5.4); 47.6% |
28.1% |
Abnormal Executive function according to 1 or more errors in CDT |
71% |
p=0.008 |
|
Quach et al., 2019, USA |
430; 76.6 (7); 68% |
42% |
MCI according to < 1,5 SD in ≥ 2 tests in NPA |
57.6% |
p<0.0001 |
|
Shirooka et al., 2017, Japan |
470;73.6 (5.2); 70% |
19.4% |
Objective Cognitive Decline according to MMSE £ 23 |
54.8% |
p=0.02 |
|
Smith et al., 2020, China, Ghana, India, Mexico, Russia, South Africa |
13623;72.3 (10.9); 54.4% |
4.5% |
MCI according to Petersen Criteria |
77% |
95% CI= [1.12-2.07] |
|
Tsutsumimoto et al., 2018, Japan |
10,202;73.7 (5.5); 51.5% |
NR |
Cognitive impairment according to score <1,5 SD in each of 3 neuropsychological tests |
66.7% |
p=0.02 |
|
Woo et al., 2017, Singapour |
385; NR; 63.9% |
27.8% |
- MCI according to MMSE score between 18 and 23 - severe cognitive impairment according to MMSE < 17 |
73.1% |
-95% CI= [1.08-3.25] - 95% CI= [0.73-2.59] |
|
Yamada et al., 2013, Japan |
31; 78.9 (7.3) ; 74.2 % |
32.2% |
- MCI according to Petersen Criteria - MMSE score - scores from 4 neuropsychological tests (a: word and b: letter fluency, c: CDT, d: TMT-A) |
73.1% |
- p=0.32 - p=0.59 - p= a:0.002; b:0.88; c:0.01; d:0.53 |
|
Yang et al., 2018, Taiwan |
1067; 76.4 (6.0); 58.9% |
15.1% |
- cognitive impairment according to CDR ≥ 2 |
71.4% |
p=0.05 |
|
Zhou et al., 2022, China |
660 ; NS (sub-group) |
21.4% |
- cognitive function as continuous variable |
71.0% |
p=0.001 |
Table 2. Prospective studies assessing the association between falls and cognitive decline
|
Authors, year, country |
N; Age*; proportion of females† |
Length of follow-up |
Fallers (%) |
Comparison |
STROBE score |
Measurement of effect size (95% CI, p) |
|
Longitudinal analysis |
||||||
|
Ge et al., 2021, USA |
6000; NR; 56.4% |
6 years |
30.1% (baseline) |
Cognitive impairment according to score < 1/15th on memory and/or executive tests; or physician diagnosis as well; or AD8 ≥ 2 |
81.8% |
p=0.04 |
|
Ma et al., 2021, China |
965;74.9 (3.7); 53.2% |
3 years |
10.6% |
Cognitive impairment according to 20% of the lowest HDS-R scores and HDS-R > 10 |
80.7% |
p=0.01 |
|
Cross-sectional analysis |
||||||
|
Adam et al., 2021, USA |
2705; 78.5 (3.2); 45% |
18 months |
18.4% |
MCI according to CDR = 0.5 and < 10th percentile in ≥ 2 tests in NPA |
84.8% |
p<0.01 |
|
Chantanachai et al., 2022, Australia |
266 ; 78.8 (NC) ; 45 |
12 months |
39.8% |
Cognitive (a: MMSE, TMT b:A and c:B, d: FAS, e: digit symbol, f: logical memory) scores among patents with MCI according to Petersen Criteria |
76.7% |
p= a:0.51, b:0.68, c:0.87, d:0.93, e:0.65, f:0.85 |
|
Davis et al., 2017, Canada |
288; 81.5 (6.5); 69% |
12 months |
58% |
- Cognition/ processing speed composite score - Cognition / working memory composite score |
75.7% |
- p<0.01 - p>0.05 |
|
Dixe et al., 2021, Portugal ‡ |
204 ; NC ; 71.1 |
12 months |
41.7% |
- Cognitive decline according to MMSE score |
58.1% |
p=0.78 |
|
Delbaere et al., 2012, Australia |
419; 77.8 (4.6); 53.9% |
12 months |
33.7% |
-MCI according to Petersen criteria - cognitive (a: executive, b: memory, c: language, d: attention composite) scores |
93.8% |
-95%CI = [0.90–2.63] -95%CI= a: [0.97-1.55], b:[0.71-1.13], c:[0.92-1.50], d:[0.81-1.28] |
|
De Vries et al., 2013, Netherlands |
1509; 75.6 (NR) ;51.8% |
12 months |
31.0% |
Cognitive impairment according to MMSE £ 24 |
72.7% |
95%CI = [0.68–1.82] |
|
Franse et al., 2017, Europe multicentric |
18596; 74.1 (NR); 55.8% |
2 years |
8.4% |
Cognitive impairment according to composite score < 1/10th |
72.7% |
p varying between > 0.05 and <0.001 depending on the country |
|
Gillain et al., 2019, Belgium |
96; 71.3 (5.4); 50% |
2 years |
36.5% |
MoCA score |
81.8% |
p=0.62 |
|
Hoffman et al., 2017, USA |
4528;76.4 (NR);58% |
24 months |
48.6% |
Cognitive impairment according to TICS £ 8 |
65.6% |
p=0.36 |
|
Makizako et al., 2013, Japan |
42; 75.6 (6.3); 42.9% |
12 months |
26.2% |
MMSE score among patients with MCI according to Petersen criteria |
57.6% |
p=0.11 |
|
Mirelman et al., 2012, Israel |
256;76.5 (4.5);61% |
5 years |
71% |
- MMSE score - a: Executive, b: attention, c: visual-spatial, d: memory composite scores |
87.9% |
-p=0.59 - p= a: 0.02, b: 0.002, c: 0.74, d:0.82 |
|
Pelaez et al., 2015, Spain ‡ |
74; 84(7); 79.7% |
20 months |
79.7% |
MMSE score |
54.5% |
95%CI = [1.03–1.40] |
|
Ranaweera et al., 2013, Sri Lanka |
1200 ; 71.4 (6.8) ; 57% |
4 months |
12.8% |
Cognitive impairment according to MMSE < 24 |
55.9% |
95%CI = [0.33–3.98] |
|
Tchalla et al., 2014, USA |
765; 78.1 (5.4); 63.8% |
5 years |
69.9% |
Cognitive impairment according to MMSE < 24 |
71.9% |
p<0.005 |
|
Ward et al., 2019, USA |
365; NR; 67% |
36 months |
NR |
(a: amnestic, b: non-amnestic et c: multiple domain) MCI according to score < 1,5 SD on 2 neuropsychological tests |
65.6% |
95%CI= a: [1.10–2.61], b: [0.31–2.69], c: [0.25–1.91] |
|
Zheng et al., 2012, Australia |
287; 77.8 (4.5); 53.7% |
12 months |
44.2% |
TMT (A-B) score |
69.6% |
p=0.03 |
Moreover, we clarified in the methods our threshold to reach the significancy, as follows: “we considered the results of the analyses to be significant if p<0.05” (line 139).
For a deeper understanding of our methods, we also detailed how we interpreted the neuropsychological tests, as follows: “If applicable, interpretation of neuropsychological tests followed authors’ recommendations. » (lines 132-133)
- The authors need to further describe the use of the terms “prospective” and “retrospective” when being applied to cross-sectional analyses. This is confusing. Typically, a study cannot be both “prospective or retrospective” and cross-sectional.
We differentiated the method of collecting the falls (which can be retrospective or prospective) and the analyses (which can be cross-sectional or longitudinal). Indeed, many so-called prospective studies only propose cross-sectional analyses (for example, at inclusion) which, in our opinion, are not of the same interpretation as longitudinal analyses. We have specified this point in the methods:” We differentiated between retrospective studies and prospective studies, i.e. depending on how fall events were recorded. Next, when considering prospective studies, we differentiated between longitudinal designs and cross-sectional designs according to the analyses performed. » (lines 134-137)
For example, Baixinho et al. conducted a retrospective study, in which fall history was collected in medical records in the previous year and data were analyzed in a cross-sectional way. In Chantanachai et al. study, falls were recorded prospectively leading to cross-sectional analyses of baseline variables between fallers and non-fallers.
- Please double check that the criteria listed under “comparison” are being consistently described in the results section.
Definitions and tests were detailed in “3.5 Definition and assessment of cognitive status” in the aim to highlight the variety of definitions used. According to your recommendation, we detailed the type of tests in the tables in the column “comparison”. We also reported in the results section as follows : “As described in the two tables, some studies included several types of patient (see below) and/or applied several types of cognitive or scores : global scores such as MMSE or MoCA, and also neuropsychological tests , i.e. clock drawing test, word recall, digit span and symbol, Trail Making test A and B, fluency, Stroop test, similarities, logical memory, which may be used to define cognitive impairment or as continuous variables. “ (lines 188-192)
We double checked the results and corrected the manuscript as follows :
- “When considering the 22 retrospective studies with cross-sectional analyses, 15 (38%) of the 39 comparisons revealed a significant association between cognition and falling. Cognitive status was significantly associated with falls in 5 (71%) of the 7 comparisons with a clear definition of MCI and 6 (60%) of 10 comparisons concerning cognitive impairment, no dementia. A global cognitive score was significantly associated with falls in 2 (25%) of the 8 studies. One of the 4 studies assessing cross-sectionally an executive score and falls (25%) found a significant association (Figure 2).” (lines 195-201)
- “Overall, a positive association was found in 23 (36%) of the 64 comparisons in cross-sectional studies » (line 217-218)
Thus, we also corrected Fig 2 :
- In the discussion section, please distinguish between MCI and falls and cognitive impairment and falls. In the results the authors report that in 71% of articles that reported a clear definition of MCI, there was a positive association between MCI and falls. This seems like a strong association, however, this is not the way it is described in the discussion. Additionally suddenly using the term mild-to-moderate cognitive impairment in the discussion is confusing.
We agree with the reviewer that this could be interpreted as a strong association. However, this result included all the studies regardless of their design. Actually, in the studies with a prospective design, only 25% of the analyses were positive and significant. This is why we chose to temper these results.
We also reported the following results in the discussion “In the present systematic review, we found a significant, positive association in only respectively 54% and 50% of the cross-sectional comparisons respectively between falling and MCI’s group and between falling and cognitive impairment, no dementia’s group. More precisely, in the included retrospective analyses with cross-sectional analyses, 5 (71%) of the 7 comparisons with a clear definition of MCI, 6 (60%) of the 10 comparisons with cognitive impairment, no dementia, and 2 (25%) of the 8 comparisons with global cognitive scores. When considering prospective studies with cross-sectional analyses, only 30% found a significant, positive association between cognition and falls: 1 (25%) of the 4 comparisons with a clear definition of MCI, 2 (33%) of 6 comparisons with cognitive impairment, no dementia, and one (25%) of the 4 comparisons with global cognitive scores. The two prospective, longitudinal studies found a causal relationship between cognitive impairment and falling but not in the two retrospective, longitudinal analyses that used specific cognitive scores. “ (lines 250-262).
To be in line with your recommendations and our manuscript, we avoid mild-to-moderate cognitive impairment in the discussion section to “non-major cognitive impairment” (lines 272-273).
Minor:
- Please do not use the acronym “MCI” in the title, use Mild Cognitive Impairment instead.
This is a typo, and we apologize for that. Finally, we chose just “cognitive impairment” to be in line with your first comment.
- Tables 1. And 2. – The inclusion of the MCI criteria is very helpful; however the column heading “comparison” is not informative and is somewhat confusing.
This column “comparison” aimed to include the type of the cognitive impairment and the type of cognitive test used in each article. We renamed it “cognitive comparator: definition”. Thus, we also corrected cognitive impairment to “cognitive impairment, no dementia” in the tables to be in line with our manuscript.
- Line 222 “ This section…” should be deleted.
My apologizes for this typo. It is deleted.
- In the discussion section “MCI status increased the risk of gait disorders and thus of falling” this needs a citation and more context. The authors did not look at this in their review, so it needs to be cited from other sources. In general, this paragraph seems out of place and the conclusions in it do not seem related to previous statements in the paragraph
Many studies showed an association between MCI and gait impairment, especially in dual-task tests or electronic treadmills record. Therefore, some authors consider MCI at risk of fall since these impairments are associated with falling. But this appears quite artificial because of the absence of strong association between fall prevalence or incidence and MCI. Our initial intention was to show the limitations of this kind of research, but we agree this section does not contribute to the discussion of our results. We removed it.
- Line 304 “Both resulted significantly”, change the phrasing of this
We rephrased it as follows: “both showed a positive and significant association between falls and cognitive impairment, no dementia” (lines 320-321).
- Line 314: “Fourthly, in order….” This sentence/ conclusion does not make sense.
Indeed. We corrected as follows : “Fourthly, we included only studies with falling outcomes and excluded indirect markers, such as gait disorders, fear of falling and standardized walking tests” (lines 331-333).
- Line 335, omit “data”, from “literature data”
It is corrected (line 351)
Reviewer 2 Report
The present study does not meet the novelty or groundbreaking data for publication. As authors well mentioned, a more prospective study should be performed, instead, with standardized data collection methods for a better understanding of the putative link between falls and MCI.
Author Response
We thank the reviewer 2 for his/her comments and appreciations. As a systematic review, our study aimed to summarize literature about falls and cognitive impairment without dementia. Thus, our first objective is to update the association between those two disorders. We showed the heterogeneity of results that failed to be a robust association. We believed our novelty is the inclusion of all the articles, even those without a clear definition of the MCI, and the comprehensive analyses of the methodology in each article. Despites the multitude of articles in this domain, we still failed to find consensus and consistent results. This is why we suggest a further longitudinal, prospective and dedicated study.
To be clearer, we change the title as follows:
“The nebulous association between cognitive impairment and falls in older adults : a systematic review of the literature”
Reviewer 3 Report
Reviewer Comment
The subject of the research was important in terms of examining both geriatrics and cognitive status and falling. Considering the aging world population, it is important and valuable to reveal the relationship between preventive approaches and cognitive status to prevent falls and the burden of cognitive status on the health system. In addition to the importance of your research, its originality and justification have not been fully revealed. It should be explained better what kind of deficiency your study fills in the literature, and on what grounds it was planned. Another shortcoming is that the research is far from homogeneous. Systematic reviews delve deeper into more goal-oriented and specific homogeneous research, linking clear and more concrete results. By including both retrospective and prospective studies, you have compared these studies somewhat superficially. And instead of examining in-depth, you couldn't examine dissimilar articles on a common denominator. Here, my criticism was a little more about the planning phase of the study. Apart from these, the manuscript was written in accordance with the systematic review rules and template and in a systematic way. I think that with the few corrections I have suggested, the manuscript will turn into a better form.
1. In the method section of the abstract, add the information that the text of the systematic review was written according to the PRISMA checklist and the quality classification of the articles used was made according to STROBE.
2. The title of the manuscript should be edited to highlight more topics. It should be added to the title that the manuscript is a systematic review. In addition, the fact that the title is in the form of a short sentence in the form of a question causes a simple appearance. A more interesting and target-oriented title should be created. The headline is the first part of an article read by the reader. Therefore it is very valuable. Edit the title with these suggestions.
3. The second and third welds are very old. The sources you provide prevalence information should be up-to-date.
4. You need to explain the reason for the research a little more clearly. While specifying the purpose, it should be stated more clearly why your work is necessary and what kind of deficiency it will fill in the literature. Towards the end of the introduction, you should emphasize the originality of your work in more sources and studies. You made little reference to literature knowledge while justifying. Make the necessary arrangements.
5. Page 2, line 64;“Furthermore, it is not clear whether cognitive impairment, no dementia is associated with the prevalence of falls.” Add source reference.
6. Page 2, line 86; “We extracted the list of publications into an online reference management tool (EndNote®, version 20, Clarivate, Philadelphia, PA, USA) to eliminate duplicates and then used a collaborative online tool (Covidence®, Melbourne, Australia) for the following steps ” Add source reference.
7. It should be added that the articles selected for the systematic review of the limitation section are not homogeneous, the research methods are different, and therefore they cannot be examined in depth by presenting the comparison and results clearly.
Author Response
We thank the reviewer3 for his/her relevant recommendation and comments. We are grateful for the opportunity to re-submit our manuscript with corrections. We have addressed each comment (in black Times New Roman font) in turn and present our responses in blue Times New Roman font.
The subject of the research was important in terms of examining both geriatrics and cognitive status and falling. Considering the aging world population, it is important and valuable to reveal the relationship between preventive approaches and cognitive status to prevent falls and the burden of cognitive status on the health system. In addition to the importance of your research, its originality and justification have not been fully revealed. It should be explained better what kind of deficiency your study fills in the literature, and on what grounds it was planned. Another shortcoming is that the research is far from homogeneous. Systematic reviews delve deeper into more goal-oriented and specific homogeneous research, linking clear and more concrete results. By including both retrospective and prospective studies, you have compared these studies somewhat superficially. And instead of examining in-depth, you couldn't examine dissimilar articles on a common denominator. Here, my criticism was a little more about the planning phase of the study. Apart from these, the manuscript was written in accordance with the systematic review rules and template and in a systematic way. I think that with the few corrections I have suggested, the manuscript will turn into a better form.
We agree that our approach is large. We decided to include all-types of studies to be as comprehensive as possible regarding the literature. There is indeed no consensus in the falling research area : recording of falls, analyses, etc.
This is certainly why we showed heterogenous results. But, according to us, this heterogeneity is a relevant result because it reflects the difficulty to answer to the question of the association between falls and cognitive impairment. This also highlighted the need for further clarification.
We detailed also further the need of this systematic review.
- In the method section of the abstract, add the information that the text of the systematic review was written according to the PRISMA checklist and the quality classification of the articles used was made according to STROBE.
We added this in the methods section :” This systematic review followed indeed the PRISMA guidelines [16], and has been registered at the PROSPERO (International Prospective Register of Systematic Reviews) (CRD42022363363).” (lines 98-100) and in the abstract, as you suggested: “According to PRISMA guidelines, we searched five electronic databases (EMBASE, Web of Science, Medline, CINAHL, and PsychINFO) for articles published between January 2011 and August 2022 on observational studies of older people with a cognitive assessment and/or cognitive impairment diagnosis and recording of falls. Their quality was reviewed according to STROBE checklist. » (lines 26-30)
- The title of the manuscript should be edited to highlight more topics. It should be added to the title that the manuscript is a systematic review. In addition, the fact that the title is in the form of a short sentence in the form of a question causes a simple appearance. A more interesting and target-oriented title should be created. The headline is the first part of an article read by the reader. Therefore it is very valuable. Edit the title with these suggestions.
We suggested the following title : “The nebulous association between cognitive impairment and falls in older adults : a systematic review of the literature”
- The second and third welds are very old. The sources you provide prevalence information should be up-to-date.
Epidemiology references have been updated :
- Luebbert, S.; Christensen, W.; Finkel, C.; Worsowicz, G. Falls in Senior Adults: Demographics, Cost, Risk Stratification, and Evaluation. Mo. Med. 2022, 119, 158–163.
- Kramarow, E. Deaths From Unintentional Injury Among Adults Aged 65 and Over: United States, 2000–2013. NCHS Data Brief 2015, 9.
- Kenny, R.A.; Romero-Ortuno, R.; Kumar, P. Falls in Older Adults. Medicine (Baltimore) 2017, 45, 28–33, doi:10.1016/j.mpmed.2016.10.007.
- Burns, E.R.; Stevens, J.A.; Lee, R. The Direct Costs of Fatal and Non-Fatal Falls among Older Adults - United States. J. Safety Res. 2016, 58, 99–103, doi:10.1016/j.jsr.2016.05.001.
- Zhang, L.; Wang, J.; Dove, A.; Yang, W.; Qi, X.; Xu, W. Injurious Falls before, during and after Dementia Diagnosis: A Population-Based Study. Age Ageing 2022, 51, afac299, doi:10.1093/ageing/afac299.
- You need to explain the reason for the research a little more clearly. While specifying the purpose, it should be stated more clearly why your work is necessary and what kind of deficiency it will fill in the literature. Towards the end of the introduction, you should emphasize the originality of your work in more sources and studies. You made little reference to literature knowledge while justifying. Make the necessary arrangements.
We added the following justifications in the introduction : “In a systematic review published in 2012 [16], Muir et al. assessed 26 prospective cohort studies of institutionalized or community-dwelling people aged 60 and over, with falls as the main outcome and a cognitive assessment at inclusion. The researchers concluded that the global cognitive decline (i.e. in all stages) was associated with falls in general and with serious falls in non-institutionalized adults. More specifically, impaired executive function was associated with a greater risk of falling. But early stages of cognitive impairment were not specifically compared.
Since 2012, additional data on the potentially causal association between the risk of falling and cognitive impairment, no dementia have been published, especially in patients with MCI. If it is strongly established that patients with non-major cognitive impairment, such as cognitive impairment, no dementia and/or MCI, have more gait disorders than cognitively unimpaired counterparts, their association with falling remains unclear. When compared, the prevalence of falls is indeed not significantly higher in these patients, but this has rarely been the focus of studies where this was the main objective. “ (lines 68-81 )
The following references have been also added in the aim to illustrate the lack in the literature about falling and non-major cognitive impairment : “
- Liu-Ambrose, T.; Ashe, M.C.; Graf, P.; Beattie, B.L.; Khan, K.M. Mild Cognitive Impairment Increases Falls Risk in Older Community-Dwelling Women. Phys. Ther. 2008, 88, 1482–1491, doi:10.2522/ptj.20080117.
- Allali, G.; Launay, C.P.; Blumen, H.M.; Callisaya, M.L.; De Cock, A.-M.; Kressig, R.W.; Srikanth, V.; Steinmetz, J.-P.; Verghese, J.; Beauchet, O. Falls, Cognitive Impairment, and Gait Performance: Results From the GOOD Initiative. J. Am. Med. Dir. Assoc. 2017, 18, 335–340, doi:10.1016/j.jamda.2016.10.008.
- Page 2, line 64;“Furthermore, it is not clear whether cognitive impairment, no dementia is associated with the prevalence of falls.” Add source reference.
Please see our answer above.
- Page 2, line 86; “We extracted the list of publications into an online reference management tool (EndNote®, version 20, Clarivate, Philadelphia, PA, USA) to eliminate duplicates and then used a collaborative online tool (Covidence®, Melbourne, Australia) for the following steps ” Add source reference
We specified the accordance with PRISMA guidelines and the registration in PROSPERO register : “This systematic review followed indeed the PRISMA guidelines [21], and has been registered at the PROSPERO (International Prospective Register of Systematic Reviews) (CRD42022363363). » (lines 98-100)
- It should be added that the articles selected for the systematic review of the limitation section are not homogeneous, the research methods are different, and therefore they cannot be examined in depth by presenting the comparison and results clearly.
We agree with your comment. But, as precised above, this heterogeneity is less a limitation than a result in our systematic review. We decided indeed to include all-types of methods.
In the limitation section, we removed the following sentence “The included studies did not all have the same design and so may be difficult to compare » to detail it in a further point :
“Lastly, if the included studies varied greatly according to our inclusion criteria, it also makes their comparison and presentation more complex” (line 348-349)
Round 2
Reviewer 1 Report
The authors have been very responsive to reviewer comments. The manuscript is much clearer and has a more appropriate definition of cognitive impairment. I recommend a minor review for typos/spell check, and English language.
Reviewer 2 Report
The present review lack novelty due to the heterogeneity of results and little significance in terms of groundbreaking data for publication.